# CRISPR/Cas9-Mediated Genome Editing in Cancer Therapy

**DOI:** 10.3390/ijms242216325

**Published:** 2023-11-15

**Authors:** Shuai Ding, Jinfeng Liu, Xin Han, Mengfan Tang

**Affiliations:** 1Department of Biochemistry and Molecular Biology, School of Medicine and Holistic Integrative Medicine, Nanjing University of Chinese Medicine, Nanjing 210023, China; shuaiding@njucm.edu.cn; 2Department of Immunology, School of Medicine and Holistic Integrative Medicine, Nanjing University of Chinese Medicine, Nanjing 210023, China; liujinfeng@njucm.edu.cn

**Keywords:** CRISPR/Cas9, genome editing, CAR-T therapy, CRISPR screening, cancer model

## Abstract

The Clustered Regularly Interspaced Short Palindromic Repeats/CRISPR-associated protein 9 (CRISPR/Cas9) system, an RNA-based adaptive immune system found in bacteria and archaea, has catalyzed the development and application of a new generation of gene editing tools. Numerous studies have shown that this system can precisely target a wide range of human genes, including those associated with diseases such as cancer. In cancer research, the intricate genetic mutations in tumors have promoted extensive utilization of the CRISPR/Cas9 system due to its efficient and accurate gene editing capabilities. This includes improvements in Chimeric Antigen Receptor (CAR)-T-cell therapy, the establishment of tumor models, and gene and drug target screening. Such progress has propelled the investigation of cancer molecular mechanisms and the advancement of precision medicine. However, the therapeutic potential of genome editing remains underexplored, and lingering challenges could elevate the risk of additional genetic mutations. Here, we elucidate the fundamental principles of CRISPR/Cas9 gene editing and its practical applications in tumor research. We also briefly discuss the primary challenges faced by CRISPR technology and existing solutions, intending to enhance the efficacy of this gene editing therapy and shed light on the underlying mechanisms of tumors.

## 1. Introduction

Recently, the morbidity and mortality rates of cancer have been increasing rapidly, posing a significant threat to human health. Cancer, a refractory and multifaceted disease, fundamentally originates from a cumulative series of mutations in the cellular genome and epigenome. These mutations activate oncogenes and deactivate tumor suppressors, leading to sustained proliferative signaling, genome instability, and evasion of growth suppressors [1]. As mutant genes continue to accumulate, they eventually give rise to tumorigenesis, altering metabolism, cell structure, and motility [2]. Conventional cancer therapies such as surgery, radiotherapy, and chemotherapy, while widely employed worldwide, are limited in their complexity when dealing with tumors. Consequently, there is an urgent need for novel cancer therapy strategies.

Meanwhile, genome editing tools have demonstrated great promise in cancer treatment, offering significant advantages by targeting oncogenes and tumor suppressor genes to control tumor growth and progression. These tools include zinc finger endonuclease (ZFN) [3], transcription activator-like effector nuclease (TALEN) [4], and the clustered regularly interspaced short palindromic repeats/CRISPR-associated nuclease9 (CRISPR/Cas9) system [5]. Among these, the CRISPR/Cas9 system, as a third-generation gene editing tool, exhibits several advantages over ZFN and TALEN, such as high efficiency, simple design, rapid operation, and low cost.

Initially, scientists observed that many viruses in the wild environment posed threats to the survival of bacteria and archaea [6]. The CRISPR/Cas9 system, functioning as an adaptive immune system defense, is found across a wide range of bacterial species [7]. This system comprises the Cas9 nuclease, CRISPR RNA (crRNA), and trans-activating crRNA (tracrRNA). The crRNA is initially transcribed as pre-crRNA, which, upon binding with tracrRNA and the action of RNase III, is processed into mature crRNA. TracrRNA is transcribed separately and then binds to mature crRNA through base complementary pairing, undergoes RNase III trimming, and associates with Cas9 to form a complex capable of DNA cleavage, combating viruses and plasmids. The crRNA contains nucleotide sequences complementary to the target DNA, specifying the cleavage site for Cas9 nuclease, while tracrRNA acts as a structural scaffold aiding the binding of crRNA to Cas9 nuclease [8].

Subsequently, for gene editing applications, scientists designed a single-guide RNA (sgRNA), which combines the functions of crRNA and tracrRNA into a single RNA molecule [9]. The sgRNA binds with the Cas9 nuclease, guiding it to specific DNA sequences, and induces double-strand breaks (DSBs). The Cas9 endonuclease is responsible for initiating these DSBs. Following this, cellular repair mechanisms, such as non-homologous end joining (NHEJ) or homology-directed repair (HDR) [10], are activated to repair or modify these breaks, thereby achieving genome editing [5]. While various categories of CRISPR and CRISPR-associated (Cas) proteins have been discovered, CRISPR/Cas9 stands out as the most widely utilized and extensively researched. This system is rapidly advancing and capturing the interest of researchers across diverse fields [11], particularly in cancer research [12]. In recent years, the CRISPR/Cas9 system has yielded impressive outcomes in cancer research, addressing various types, such as liver cancer, breast cancer, and colorectal cancer, among others [13,14,15,16]. These achievements have contributed significantly to our comprehensive understanding of cancer initiation and progression.

Nonetheless, several unresolved obstacles, including off-target mutations, must be addressed to enhance its effectiveness and facilitate clinical applications. In this review, we aim to shed light on the potential of the CRISPR/Cas9 system in contemporary cancer research and its applications in diagnosis and treatment. Furthermore, we will discuss the challenges and constraints associated with the clinical utilization of the CRISPR/Cas9 system.

## 2. CRISPR/Cas System Mechanism

The CRISPR/Cas system, which originally evolved as a bacterial immune system to defend against infections and plasmid transfers in nature [6], has been ingeniously repurposed by scientists into a sophisticated genetic editing tool. It can be categorized into six main types and two main classes: class I (types I, III, and IV) and class II (types II, V, and VI) [17] (Figure 1), based on the CRISPR/Cas loci classification [18]. Type I, II, and V systems recognize and cleave DNA, while type VI edits RNA, and type III edits both DNA and RNA. Each system possesses unique attributes, including varying protospacer adjacent motif (PAM) regions, diverse Cas protein dimensions, and different cleavage sites.

Class I CRISPR/Cas systems consist of multi-subunit Cas-protein complexes, while class II systems rely on single Cas proteins. The primary utilization of the type II CRISPR system involves facilitating genome editing within eukaryotic cells to target specific DNA sequences [19]. This includes Cas1, Cas2, Cas4, and Cas9 nuclease. The relatively straightforward structure of type II CRISPR/Cas9 has enabled its thorough examination and widespread application in gene editing. The Cas9 nuclease, derived from Streptococcus pyogenes (SpCas9), is the signature protein of the type Ⅱ CRISPR system. The CRISPR/Cas9 system comprises the Cas9 nuclease and a guide RNA (gRNA). Typically, the gRNA consists of crRNA and transactivating crRNA (tracrRNA), which base-pairs with DNA target sequences, enabling Cas9 to introduce site-specific double-strand breaks (DSBs) into DNA. CrRNA and tracrRNA are often synthesized into a single sgRNA to simplify the design of target-specific cleavage sites [5]. The Cas9 nuclease consists of the recognition (REC) lobe and the nuclease (NUC) lobe. The REC lobe includes the REC1 and REC2 domains, which are responsible for binding guide RNA. The NUC lobe comprises the RuvC-, HNH-, and PAM-interacting domains. The RuvC and HNH domains facilitate the cleavage of single-stranded DNA, including DSBs about three base pairs upstream of PAM (5′-NGG-3′) [19,20]. Subsequently, DSBs induce cellular repair mechanisms, including canonical non-homologous end joining (c-NHEJ/NHEJ), alternative end joining, or microhomology-mediated end joining (alt-EJ/MMEJ) and homology-directed repair (HDR) [10] (Figure 2). NHEJ and MMEJ often result in gene insertions or deletions (INDELs), disrupting protein-coding sequences and establishing functional gene knockouts. The repair template includes the target gene and homologous sequences of the target sequence (homology arms). HDR can be used to insert specific genes at cleavage sites by introducing donor DNA templates [18].

## 3. CRISPR/Cas9 in CAR-T-Cell Therapies

Among the various anticancer strategies, immunotherapy stands out as a treatment that harnesses the immune system of the body to combat cancer. Immunotherapy has the potential to provide more potent and long-lasting treatment outcomes compared with conventional approaches.

CAR-T cells, a type of immunotherapy, represent a significant milestone in personalized cancer treatment. They have emerged as a potent therapeutic option for patients with cancers (Figure 3), such as acute lymphoblastic leukemia (ALL), chronic lymphocytic leukemia (CLL), lymphoma, and multiple myeloma (MM) [21,22]. The United States Food and Drug Administration (FDA) has approved multiple CAR-T-cell products for large B-cell lymphoma (LBCL), B-cell acute lymphoblastic leukemia (B-ALL), follicular lymphoma, and mantle cell lymphoma [23]. Furthermore, CAR-T-cell research has shown promise in addressing solid tumors such as glioma, gastric cancer, non-small-cell lung cancer, and liver cancer [24,25,26,27].

CAR-T cells are genetically modified T cells designed to express CARs. These synthetic proteins guide these immune cells to tumor antigens, allowing them to specifically identify and attach to proteins on the surface of tumor cells. This process activates T cells and enhance their antitumor activity, ultimately leading to the elimination of cells expressing the antigen [28].

CARs exhibit a modular structure that includes an antigen-binding domain, a hinge region, a transmembrane domain, a co-stimulatory domain, and an activation domain. This structure combines elements from both the T-cell receptor (TCR complex) and antibody (Abs) structures. The T-cell receptor recognizes antigens (Ags) bound to the major histocompatibility complex (MHC). The CAR structure can be further divided into ectodomain, transmembrane domain and endodomain, domains. The antigen-binding domain consists of a single-chain variable fragment (scFv) linked to the endodomain by a spacer and transmembrane layer in the ectodomain. This scFv fragment comprises the variable parts of the light chain (VL) and the heavy chain (VH), connected by a short polypeptide linker [29]. The scFv fragment can bind specifically to antigens on the tumor cell surface. The endodomain components include signaling modules (*CD3ζ*) and co-stimulatory molecules that confer specific functions on the receptor-expressing cell. Incorporating co-stimulatory domains like *CD28*, *4-1BB*, or *OX40* enhances the in vivo cell proliferation, survival, and antitumor activity of T cells bearing these receptors. CARs are categorized into different generations based on the number of co-stimulatory domains. The intracellular signaling domain typically includes a T-cell activation domain derived from the CD3ζ chain of the T-cell receptor and co-stimulatory domains, often containing regions with immunoreceptor tyrosine-based activation motifs found in CD28 or 4-1BB [29].

The first-generation CAR involves only the intracellular signaling domain CD3ζ, while the second-generation CAR includes an additional co-stimulatory molecule alongside *CD3ζ*. The third-generation CAR includes another co-stimulatory domain. Recently developed fourth-generation CAR-T cells can effectively stimulate downstream transcription factors and trigger cytokine release upon detecting tumor-associated antigens (TAAs) bound to CARs. Notably, the fifth generation of CARs, based on the design of the second generation, utilizes gene editing to suppress the expression of the T-cell receptor (TCR) gene (TRAC), aiding in the elimination of TCR α and β chains.

Despite the clinical success of CAR-T therapy for hematological malignancies, several constraints continue to hinder its widespread adoption. Firstly, the current conventional therapy involves labor-intensive and time-consuming allogeneic adoptive cell transfer, leading to high costs and extended treatment durations. Secondly, challenges arise in cases involving neonatal and elderly patients, where obtaining sufficient high-quality T cells for personalized CAR-T-cell production often proves difficult [30]. This issue is exacerbated in patients undergoing chemotherapy or radiation therapy, which can result in reduced quantities and compromised quality of autologous T cells. Additionally, the diverse expression of tumor antigens and the immune evasion strategies employed by tumor cells highlight the need for CAR-T cells capable of effectively targeting multiple tumor antigens simultaneously. Adding to the complexity, the immunosuppressive tumor microenvironment (TME) disrupts T-cell activity, leading to T-cell differentiation and depletion. Alongside the limited antitumor potential of these cells, a significant drawback associated with their use is the manifestation of side effects. Notably, there are further limitations, including T-cell dysfunction, cellular toxicity, resistance to transforming growth factor-beta (TGF-β), risks of graft-versus-host disease (GvHD), cytokine release syndrome (CRS), restricted migration, and limited tumor infiltration [31,32,33]. Within the realm of CAR-T-cell therapy, it is crucial to distinguish between two modalities: autologous therapy, which utilizes the patient’s own T cells, and allogeneic therapy, which employs T cells from universal donors. We have thoroughly compared these two approaches in Table 1, outlining their respective advantages and limitations. This comparison is intended to provide clinicians and researchers with a valuable reference to better understand the practical implications of each treatment modality and to make the most appropriate decisions for patient-tailored therapeutic strategies. Hence, the development of allogeneic T cells with enhanced antitumor effects is of paramount importance. The application of gene editing technologies has revolutionized the field of CAR-T-cell therapy. By enabling precise alterations within the genome, researchers can enhance the specificity, functionality, and persistence of CAR-T cells, paving the way for more effective treatments. To depict these advancements, Figure 4 will provide a detailed illustration of CRISPR-modified CAR-T cells, showcasing the targeted integration of genetic modifications that bolster antitumor responses.

The CRISPR/Cas9 system offers a promising avenue for enhancing the anticancer response of CAR-T cells through genetic engineering [34]. Traditional T-cell therapies often result in T-cell dysfunction or exhaustion. To enhance the performance and longevity of engineered T cells, Stadtmauer et al. [35] conducted ablation of two genes encoding the endogenous T-cell receptor (TCR) chains, namely TCRα (TRAC) and TCRβ (TRBC). These genes play crucial roles in the assembly and expression of the T-cell receptor complex. By eliminating them, the natural T-cell receptor is effectively removed, enabling T cells to exclusively recognize and target cancer cells through the synthetic CAR receptor. This not only reduces the risk of T cells erroneously attacking healthy cells but also decreases the likelihood of TCR mismatching. Additionally, they introduced cancer-specific TCR transgenes, such as NY-ESO-1, using CRISPR technology to enhance the recognition of tumor cells. Furthermore, researchers employed the CRISPR technology to delete the gene encoding the programmed cell death protein 1 (PD-1; PDCD1). This intervention significantly bolstered the antitumor efficacy of CAR-T cells against xenograft tumor transplantation. In subsequent clinical trials, although chromosomal translocation phenomena were detected, their frequency gradually decreased over time. CRISPR-edited T cells persisted in the body for up to nine months, indicating minimal immunogenicity under these conditions and confirming the feasibility and safety of multiple CRISPR gene editing techniques in CAR-T immunotherapy. In the treatment of hematologic malignancies, CAR-T-cell therapy has achieved significant success. However, resistance issues limit its widespread application. Through high-throughput CRISPR/Cas9 screening, Yan et al. [36] identified and validated the critical role of CD58 in immune evasion during CAR-T-cell therapy. CD58 is a ligand for the T-cell co-stimulatory molecule CD2 and is frequently mutated or downregulated in hematologic tumors. The loss of *CD58* in tumor cells results in the formation of suboptimal immunological synapses (ISs), thereby attenuating CAR-T-cell functions, including cell proliferation, degranulation, cytokine secretion, and cytotoxicity. Yet, strategies to overcome CAR-T-cell therapy resistance due to CD58 loss remain to be developed. CAR-T-cell therapy in clinical trials predominantly involves autologous engineered T cells, but this approach presents challenges related to T-cell quality and quantity, as well as prolonged and cost-intensive manufacturing processes. To circumvent these limitations, Ren et al. [37] employed the CRISPR/Cas9 system to simultaneously knockout multiple genomic loci, resulting in the creation of universal CAR-T cells lacking endogenous T-cell receptor (TCR) and HLA class I (HLA-I) expression. These biallelic-deficient T cells show reduced sensitivity to allogeneic responses, thus preventing graft-versus-host disease. Furthermore, the enhanced therapeutic efficacy of gene-edited allogeneic CAR-T cells in tumor models was achieved via CRISPR-mediated deletion of endogenous PD1. Nonetheless, it is important to acknowledge that these triple gene-edited CAR-T or TCR T cells may trigger NK cell activation, potentially leading to the eventual rejection of the edited T cells. Additionally, activated T cells show elevated expression of HLA class II, which could expedite the rejection of infused allogeneic T cells. Consequently, future research endeavors should focus on addressing these potential issues to ensure the successful clinical application of edited CAR-T cells.

In the development of CAR-T-cell therapies, the application of CRISPR/Cas9 technology represents a significant breakthrough. Nevertheless, the Cas9 protein, which is derived from bacteria, may elicit an immune response upon introduction into the human body due to its exogenous nature. Such responses could lead to the generation of antibodies and T cells against Cas9, accelerating the degradation of the protein and compromising its gene-editing capabilities [38]. The risk of this immune reaction is particularly pronounced in therapies requiring prolonged expression of Cas9, where sustained protein presence could invoke a robust immunological memory. To mitigate these immunogenic risks, researchers are devising strategies such as the deployment of modified Cas9 variants that minimize immunogenic epitopes to reduce detectability by the immune system [39]. Furthermore, transient expression systems and non-inflammatory vectors, like adeno-associated viruses (AAVs) [40], are being employed to further diminish immune activation, thus enhancing the safety and efficacy of the therapy.

Several ongoing clinical trials are actively recruiting participants to demonstrate the potential of CRISPR technology before introducing CAR-T-cell products for the clinical management of malignancies. These trials predominantly fall under phase one and are primarily being conducted in China and the United States. Table 2 provides details on each clinical trial, such as the CAR-T-cell variant, clinical stage, participant count, geographical location, and trial identification number.

Certainly, in addition to CAR-Tcell therapy, there are several other CAR-based immunotherapies currently under active development, including CAR-M, CAR-NK cells, CAR-γδT, and CAR-NKT therapies. Each of these approaches offers distinct advantages compared with CAR-T cell therapy but also certain limitations. For example, CAR-NK cells have a finite lifespan and carry a reduced risk of off-target effects on non-tumor cells. However, this necessitates continuous administration to sustain their tumor-killing capabilities [41]. CAR-M therapy capitalizes on the abundant infiltration of macrophages in the TME, enabling them to perform phagocytic functions and more within the TME [42]. These novel therapeutic modalities serve as complementary strategies to CAR-T cell therapy, contributing to the ongoing refinement of CAR-based immunotherapies. However, it is worth noting that their limitations remain subjects of investigation by researchers.

## 4. CRISPR Screening in Cancer

The completion of the Human Genome Project (HGP) marked a major milestone in genomics research [43,44]. Although we now understand the composition of the human genome, our knowledge about the functions and interactions of individual genes remains limited, which impedes the exploration of human genetic diseases. Nevertheless, the development of CRISPR/Cas9 technology has opened up new avenues for studying genes, including their roles in cell proliferation, metastasis, immune evasion, and drug resistance [16,45,46,47,48]. This technology enhances our understanding of how genes influence complex phenotypes. In previous studies, we successfully employed CRISPR screening to identify potential cancer targets [49,50,51].

CRISPR screening is a high-throughput genetic screening technique that utilizes different CRISPR/Cas9 sgRNA libraries covering the entire genome to identify interesting genes based on cell survival or phenotypic changes [52,53]. There are three types of CRISPR screening: CRISPR knockout (CRISPRKO) screening, CRISPR interference (CRISPRi) screening, and CRISPR activation (CRISPRa) screening, including point mutagenesis [20,54,55], each of which is associated with specific libraries. These libraries, which are designed for gain or loss of function, have been widely used to uncover novel biological mechanisms (Figure 5).

CRISPR screening involves the synthesis of oligonucleotides containing single RNA guide sequences, which are then cloned into a lentiviral-based library. Cells expressing Cas9 or other Cas proteins are infected with lentivirus to ensure that only one copy of sgRNA integrates into each cell. These cells are then screened using biological assays. If the target gene affects cell fitness under specific conditions, cells with corresponding sgRNA will either decrease or become enriched in the population. CRISPR screens use distinct sgRNA sequences and next-generation sequencing (NGS) to detect changes in sgRNA frequency after phenotypic selection [56].

### 4.1. CRISPRKO Screening

CRISPRKO screening is a widely used negative selection screening platform that efficiently introduces precisely targeted loss-of-function mutations at specific genomic locations. It identifies depleted or reduced sgRNAs within a Cas9-expressing cell population. DSBs occur at the genomic loci of sgRNA targets, leading to indels and point mutations through non-homologous end joining (NHEJ) repair, disrupting the functionality of the targeted gene [57]. These sgRNA targets can be designed anywhere in the genome, including in non-coding regions, such as promoters and enhancers [57]. For instance, Wang et al. [58] conducted pooled in vivo CRISPR knockout (KO) screens in syngeneic triple-negative breast cancer (TNBC) mouse models and found that deleting the E3 ubiquitin ligase Cop1 in cancer cells reduced the secretion of macrophage-associated chemokines, reducing tumor macrophage infiltration, enhancing antitumor immunity, and strengthening the response to immune checkpoint blockade therapy. Targeting Cop1 and modulating chemokine secretion and tumor microenvironment macrophage infiltration improved TNBC cancer immunotherapy efficacy. Shi et al. [59] used CRISPRKO screening to explore the mechanisms of cisplatin resistance in six bladder cancer cell lines due to the unsatisfactory overall efficacy of cisplatin-based chemotherapy in bladder urothelial carcinoma (BUC). Their study revealed that the heterogeneous nuclear ribonucleoprotein U (*HNRNPU*) gene is a candidate associated with cisplatin resistance. Knocking out *HNRNPU* enhanced cisplatin sensitivity by regulating DNA damage repair genes, suggesting that targeting *HNRNPU* could offer a potential therapeutic approach for cisplatin-resistant bladder cancer. Therefore, the successful application of CRISPRKO screening in oncology confirms its unbiased potential to identify various cellular mechanisms, with the subsequent clinical translation of these findings holding significant promise for patients with cancer.

### 4.2. CRISPRi Screening

CRISPRi screening utilizes a catalytically inactive version of Cas9 (dCas9) that lacks endonucleolytic activity. This method allows for the reversible inhibition of target genes without altering the genome sequence [60]. It efficiently silences target genes to induce loss of function [54]. CRISPRi screening consists of three essential components: the PAM site, sgRNA, and the dCas9 protein. Cas9, one of the most commonly used Cas proteins, can cleave invading DNA through its RuvC and HNH nuclease domains, cleaving the non-complementary and complementary strands, respectively. To generate dCas9, a catalytically inactive mutant of Cas9^D10A/H840A^, the catalytic domains are mutated [61]. This mutation renders dCas9 devoid of endonuclease activity while still allowing it to form a complex with sgRNA. sgRNA is typically composed of 102 nucleotides. It encompasses a 20 nt target-specific complementary region, a 42 nt Cas9-binding RNA structure, and a 40 nt transcription termination sequence derived from S. pyogenes. In the CRISPRi strategy, the chosen sgRNA targeting sites generally lie within a range spanning 50 base pairs upstream to 300 base pairs downstream of the transcription start site (TSS) [5,60]. Once dCas9 forms a complex with sgRNA, it is directed to either transcriptional repression domains or transcriptional activation domains. The sgRNA binds to the target DNA through sequence homology, while the dCas9 protein associates with the adjacent PAM site. This specific interference disrupts transcriptional elongation, RNA polymerase binding, and transcription factor association, ultimately influencing gene expression [61]. In addition, adaptive applications have been developed using catalytically inactive Cas9 (dCas9) along with other regulatory co-factors like KRAB [62]. To downregulate targeted genes, CRISPR-dCas9-KRAB (CRISPRi) is employed to interfere with or silence transcription processes [63].

Genes contain multiple promoters capable of driving the expression of various transcript isoforms, some with opposing functions. However, current methods cannot determine the functions and phenotypes of individual transcript isoforms. Davies et al. [64] used CRISPRi screening to selectively target specific promoters of different genes, aiming to identify the functions of distinct transcript isoforms. Their study revealed that the transcript isoforms of the tumor suppressor *ZFHX3* have dual-sided functions, correlating with adverse patient outcomes. This finding was previously unattainable using prior research methods. This study suggests a direction for the subsequent development of inhibitors or activators targeting different transcript isoforms of the same gene. Long non-coding RNAs (lncRNAs) exhibit cell-type-specific expression and functions, making them potential therapeutic targets in cancer treatments. However, most lncRNAs remain unexplored. Liu et al. [65] used CRISPRi screening to identify 33 lncRNA loci in glioblastoma that modulated radiation sensitivity. Notably, the depletion of lncRNA Glioma Radiation Sensitizers (lncGRSs), among these hits, significantly enhanced the radiosensitivity of glioblastoma without impacting the viability and proliferation of normal cells. This approach holds promise as a widely applicable screening method that, when combined with other therapeutic modalities, can expedite clinical research.

### 4.3. CRISPRa Screening

CRISPRa screening targets promoter regions of gene loci to activate gene expression, enabling gain-of-function analysis for positive screening. This has significant implications for understanding the role of gene upregulation in cancer. By designing a suitable sgRNA library, genes are activated for transcription at their native loci through the CRISPRa system. Compared with previous methods using cDNA libraries for overexpression, CRISPRa modulates gene transcription at these loci, aiming to mirror physiological expression levels while using synthetic activators. Moreover, designing and cloning the sgRNA library is more cost-effective and straightforward. Initially, the fusion of dCas9 with VP64 and p65 activation domains achieved limited transcriptional activation. However, Tanenbaum et al. [66] harnessed the SunTag, a repeating peptide array, to construct the dCas9-SunTag-VP64 transcriptional system, achieving more robust transcriptional activation. This approach not only simplifies gene activation methods but also provides the potential for simultaneous activation of multiple genes. Chavez et al. [67] found that when P65 or Rta was fused separately with dCas9, there was an increase in transcriptional activity. However, when both P65 and Rta were tandemly fused to VP64, transcriptional activation activity significantly increased. In 2017, Zhang et al. [68] conducted research to incorporate the MS2-binding loop sequence into the sgRNA scaffold, recruiting two additional activation domains, p65 and HSF1, to the dCas9-VP64 fusion construct. This configuration gave rise to a synergistic activation mediator (SAM) complex, which again robustly facilitated transcriptional activation.

The CRISPRa screening system inherits the advantages of CRISPR knockout technology, including high efficiency, good specificity, and minimal off-target effects. While some molecular designs can be complex, the key modification involves altering the 5′ end sequence of the sgRNA when targeting different genes. This cost-effective approach is suitable for constructing comprehensive gene activation libraries. Bester et al. [69] combined computational analysis of cell line pharmacogenomic data with functional CRISPRa screening targeting both coding and non-coding genes, developing a comprehensive analytical strategy. By integrating protein-coding and non-coding sgRNA libraries, they identified novel genes involved in chemotherapeutic resistance. They revealed that transcriptional activation of GAS6-AS2 lncRNA leads to hyperactivation of the GAS6/TAM pathway, a resistance mechanism in multiple cancers. This methodology aids in identifying new resistance targets, enhancing clinical therapeutic opportunities. Wang et al. [70] developed a strategy based on CRISPRa and synthetic gene circuits for the identification of specific tumors, followed by the activation of immune-related genes. This circuit consists of two interactive modules, namely the oncogenic TF-driven CRISPRa effector (AND gate) and the corresponding p53-induced shut-off switch (NOT gate), working together to implement accurate tumor targeting through AND-NOT logic. Transcription factor (TF) dysregulation is a factor in tumor progression, and synthetic gene circuits can sense oncogenic TFs (onco-TFs) within the tumor, subsequently activating immune-related genes. However, tumor transcription factors are also active in normal cells, making it challenging to strictly distinguish between tumor cells and normal cells. Therefore, incorporating P53, the most frequently mutated gene in human cancers, into the sensor improves tumor recognition. Activation of immune-related genes only occurs when both tumor TF activation and p53 defects (AND-NOT) are simultaneously detected. While using P53 as a NOT gate still has some limitations, it represents a promising direction for CRISPRa-based tumor therapy platforms, enabling the identification and immune remodeling of specific tumors. Joung et al. [71] identified four candidate genes (*CD274* (PD-L1), *MCL1*, *JUNB*, and *B3GNT2*) across diverse cancer cells and mouse xenograft models. Their research underscores the feasibility of using CRISPRa to uncover candidate genes in various biological processes, enhancing researchers’ understanding of different biological phenomena.

Given the increasingly widespread adoption and significant impact of CRISPR technology in cancer research, we have created a table to concisely and comprehensively summarize numerous research findings. Table 3 offers a comprehensive overview of key data points from CRISPR screening studies, categorized by cancer type, screening objectives, specific CRISPR screening methods, experimental settings (in vitro or in vivo), candidate genes of interest, and relevant literature references. Our aim is to provide readers with a rapid and comprehensive understanding of the diversity of CRISPR screening in cancer. This table encapsulates the research pertaining to CRISPR screening discussed in this review.

## 5. CRISPR/Cas9 in a Cancer Model

Gene editing, a widely acknowledged approach, holds significant potential for treating diseases caused by genetic mutations. The advent of the CRISPR/Cas9 system has significantly enhanced the accuracy and efficiency of gene editing, enabling precise modifications at any genomic position. Over the past decade, CRISPR/Cas9, coupled with high-throughput screening technologies, has identified numerous mutated genes associated with tumor initiation and progression. These genes have found diverse applications, such as investigating cancer-related genes functions, establishing animal models for tumors, identifying drug targets, and discovering novel tumor biomarkers. However, validating these genes in animal tumor models is imperative to confirm their viability as targets, representing an initial step toward advancing precise clinical treatments. Subsequently, the development of related drugs can further boost clinical therapy success rates. The establishment of cancer models accelerates the understanding of the researchers regarding tumor development and promotes the translation of targeted therapies into clinical practice. The CRISPR/Cas9 system has become an indispensable tool for advancing clinical translation in cancer research.

Cancer research necessitates rapid, feasible tumor models that faithfully replicate the complex molecular events driving tumor progression at the cellular level.

### 5.1. In Vitro

The CRISPR/Cas9 system has emerged as a valuable tool for developing accessible and practical mouse tumor models, aiding in the identification of genes involved in cancer development. In vitro, cell lines have significantly facilitated the investigation of intricate mechanisms underlying tumor initiation, progression, and signaling transduction in cancer research. The utilization of the CRISPR/Cas9 system has expanded the capabilities of these in vitro models. For instance, Shalem et al. [57] employed a genome-scale CRISPR-Cas9 knockout (GeCKO) library with 64,751 unique guide sequences to identify essential genes in cancer and pluripotent stem cells. They subsequently screened a melanoma model to identify genes associated with resistance to RAF inhibitors, revealing novel hits including *NF2*, *CUL3*, *TADA2B*, and *TADA1*. Glioblastoma (GBM), a highly malignant tumor, remains poorly understood at the molecular level. Tejero et al. [72] employed the CRISPR/Cas9 system to introduce an inducible histone2B-GFP (iH2B-GFP) reporter gene into patient-derived GBM cells. This allowed them to track cell division history and discover potential niche factors such as hypoxia and TGFβ signaling that promote quiescence in GBM. Breast cancer exhibits histological and functional diversity, reflecting distinct genetic alterations in different subtypes. Dekkers et al. [73] utilized the CRISPR/Cas9 system to knockout four tumor suppressor genes associated with breast cancer, including P53, *PTEN*, *RB1*, and *NF1*, in human-derived breast progenitor cells. These modified cells were transplanted into mice to establish a model of estrogen-receptor-positive luminal tumors. Among these mutations, three specific mutation lineages exhibited the capacity for long-term culturing. Moreover, organoid tissues from these mutation-carrying lineages responded to endocrine therapy or chemotherapy, highlighting the successful application of CRISPR technology in constructing tumor models and enhancing our understanding of breast cancer subtypes. Zhao et al. [74] also employed CRISPR/Cas9 technology to generate *BRCA1* gene knockdown adipose-derived stem cells, aiming to investigate their interactions with breast cancer cell lines. Numerous researchers have utilized CRISPR/Cas9-engineered in vitro tumor cell systems to study the tumorigenesis process. These efforts have showcased the broader applications of CRISPR technology in oncology, fostering a deeper comprehension of tumor biology and ultimately benefiting a multitude of patients.

### 5.2. In Vivo

In summary, the construction of in vitro cell models can assist researchers in understanding tumors. However, the TME includes various other cellular components, such as immune cells and extracellular matrix cells, which can influence responses to tumor treatment [75,76]. Solely relying on the establishment of in vitro tumor models cannot fully represent the true characteristics of tumors within the body. In such cases, the creation of murine in vivo tumor models is crucial, as they can further reflect the therapeutic responses of drugs within the body. The establishment of in vitro cell models aids researchers in understanding tumors. Mouse tumor models can be broadly categorized into spontaneous tumor mouse models, induced tumor mouse models, genetically modified mouse tumor models, xenograft models, and syngeneic mouse tumor models (CDA or MDA) [77,78]. Among these, the subcutaneous (s.c.) transplantation tumor model, within the transplantation models, is commonly used for preclinical studies [79].

The subcutaneous (s.c.) transplantation tumor model involves implanting tumor cells via subcutaneous injection into immunodeficient mice for growth. It offers advantages such as a low cost, short experimental cycles, ease of operation, and good repeatability [80]. This transplantation model can be used for tumor studies involving gene knockout/overexpression using CRISPR technology. By directly transplanting cells under the skin of immunodeficient mice, it allows the assessment of tumor occurrence, growth, and drug resistance. Compared with cell culture models, this model can more accurately simulate tumor proliferation, aiding in the discovery of new targets. However, since this transplantation model is typically established through subcutaneous injection of tumor cells, it is disconnected from the original tumor microenvironment at the primary site. Its development and progression significantly deviate from the actual disease onset, and the model exhibits poor responsiveness to drugs, a low metastasis occurrence rate, and misalignment of survival data with clinical outcomes. Especially in recent years, with the increasing popularity of immunotherapy, the subcutaneous transplantation tumor model has become increasingly inadequate to meet real-world needs. As a result, the rapidly developing orthotopic transplantation tumor model, which closely mimics human physiology and pharmacology, has gained significant traction.

Compared with the subcutaneous tumor model, the orthotopic tumor model involves implantation or tissue grafting at physiologically relevant sites, providing a more faithful emulation of tumor cell growth within the TME, eliciting stronger antitumor immune responses. Erstad et al. [81], while studying pancreatic ductal adenocarcinoma (PDAC), discovered significant genetic and molecular differences between orthotopic and heterotopic tumor models when testing various surgical methods and immune-active PDAC syngeneic xenograft models. The tumor microenvironment critically influences the tumor’s response to drugs, and the orthotopic model more accurately represents the authentic features of the tumor’s primary site, including connective tissue proliferation responses and angiogenesis. Despite the longer duration required to establish orthotopic tumor models, they allow for growth within an authentic tumor microenvironment and facilitate corresponding research [82]. The orthotopic tumor model plays a significant role in preclinical research, enabling the evaluation of the antitumor effects of test compounds/drugs.

Tumor cell lines cultured in vitro can exhibit distinct phenotypes compared with their parental tumors, leading to low similarity. This significant limitation impedes the ability of the aforementioned tumor models to adequately capture tumor heterogeneity [83]. Patient-derived xenograft (PDX) models involve implanting tumor cells or tissues of the patient into immunodeficient mice for preclinical modeling. Currently, the development of patient-derived orthotopic xenograft (PDOX) models, in particular, has played a significant role in cancer biology research, drug target screening, and personalized therapy [84]. Combining CRISPR screening with PDX models offers an attractive approach to discovering individualized cancer targets. Bahrami et al. [85] established two PDX models for acute leukemias and employed a combination of ultra-sensitive proteomics and CRISPR-Cas9 reverse genetics in vivo screens. This integrated approach aimed to identify regulators of leukemia maintenance within the protective bone marrow (BM) microenvironment, potentially serving as novel therapeutic targets. Their research successfully identified ADAM10 (A disintegrin and metalloproteinase domain-containing protein 10) as a crucial regulator of PDX maintenance in vivo. Moreover, they demonstrated that targeting ADAM10 increased the sensitivity of acute leukemia cells to treatment. Their study also employed CRISPR screening to identify and validate factors contributing to PDX model stability, thereby extending the applications of PDX in preclinical drug target development and facilitating the implementation of personalized treatment strategies. In another leukemia study, Wright et al. [86] combined CRISPR screens with PDX models to investigate the mechanisms underlying the response and resistance to Bromo- and Extra-Terminal Domain Inhibitors (BETi), which have demonstrated therapeutic efficacy across various cancers. They conducted genome-wide functional loss CRISPR screening in KMT2A-rearranged (KMT2A-r) cell lines treated with BETi and identified significant BETi resistance associated with the absence of the Speckle-Type POZ Protein (*SPOP*) gene, a member of the Broad Complex, Tramtrack, and Bric-a-Brac/Poxvirus and Zinc finger (*BTB*/*POZ*) family. The validity of their findings was confirmed through in vivo PDX models, leading to potential clinical combined therapeutic strategies. Esophageal squamous cell carcinoma (ESCC), a globally prevalent form of esophageal cancer, is characterized by its propensity for metastasis, contributing significantly to its high mortality rates. Due to the lack of effective experimental models, the therapeutic strategies available for ESCC are limited in number. Xu et al. [87] employed whole-genome CRISPR/Cas9 screening coupled with genetic analyses using a highly invasive and metastatic ESCC subset and (PDX models. Their investigation revealed that elevated MEST expression in ESCC is associated with reduced patient survival and promotes tumor invasion and metastasis. Furthermore, they identified small-molecule inhibitors through a small-molecule library screening approach.

The experimental findings from these studies consistently yield exciting results, providing ample evidence that PDX models, when compared with other tumor transplantation models, offer a superior representation of clinical tumor progression and selected characteristics. These preliminary findings have illuminated the path for subsequent clinical drug development. Despite the existing limitations of PDX models, they retain irreplaceable advantages when contrasted with alternative models. In conclusion, the integration of PDX models with CRISPR-related technologies plays a pivotal role in exploring the mechanisms of tumor initiation and progression, identifying drug targets, and investigating immunotherapies, underscoring their significance in cancer research.

## 6. Limitations of CRISPR/Cas9

While CRISPR/Cas9 gene editing technology holds great promise, several limitations and risks must be addressed before its clinical application. One significant limitation is the issue of off-target effects, which remains a major concern for the CRISPR/Cas9 system [88]. While well-designed single-guide RNAs (sgRNAs) are intended to induce INDELs at specific DNA sequences, they can also occasionally induce off-target mutations at regions similar to the target sequence. These off-target mutations have the potential to disrupt normal genomic sequences, leading to compromised cellular functions and a high risk of cell death. Overcoming off-target effects is crucial for the clinical translation of CRISPR/Cas9. To mitigate off-target effects, careful attention must be paid to the specificity of sgRNAs during their design. Studies have shown that truncating the 3′ end of the sgRNA or shortening the region complementary to the target by up to three nucleotides at the 5′ end of the sgRNA can enhance target specificity [89]. Additionally, adding two guanine nucleotides to the 5′ end of sgRNAs has been shown to improve target specificity. Another avenue to explore is reducing the activity of the Cas9 enzyme, as higher Cas9 activity often results in more non-specific cleavage, which contributes to off-target effects [90]. In conclusion, addressing the challenge of off-target effects is crucial for harnessing the full potential of CRISPR/Cas9 in clinical applications. By carefully designing sgRNAs, optimizing Cas9 activity, and exploring other strategies, researchers aim to minimize off-target effects and improve the safety and precision of CRISPR/Cas9-based therapies.

Meanwhile, there are limitations in the delivery methods of the CRISPR/Cas9 system within target cells, and addressing this challenge is a critical aspect of current research. Ensuring the appropriate, safe, and precise delivery of the CRISPR/Cas9 system to tumor sites is a key focus. Currently available delivery methods include viral vectors, extracellular vesicles, nanoparticles, and exosome-based approaches [91]. However, each method comes with its own limitations. For instance, viral-vector-based delivery methods may lead to cell toxicity, immune reactions, mutagenesis, and off-target risks. The utilization of viral vectors for delivery can raise concerns due to the potential for adverse effects associated with the use of viral particles. Immune responses against the viral vectors themselves, as well as integration of the vector DNA into the host genome, are important considerations. Additionally, the potential for mutations and off-target effects resulting from the integration process could impact the safety and efficacy of the therapeutic approach [92]. Researchers are exploring lipid-based nanoparticles as a safer and more effective delivery approach [93,94].

An inherent challenge with CRISPR editing is the frequent generation of DSBs in DNA, which poses a potential risk for carcinogenesis. Specifically, when Cas9 induces DSBs in human pluripotent stem cells, it can activate the p53 pathway. This activation often leads to apoptosis in targeted cells. A concerning outcome of this is the potential enrichment of cells harboring p53 mutations in the surviving cell population, thereby escalating the risk of oncogenesis [95]. Overexpression of the *P53DD* gene or transient inhibition of the P53 DSB response using P53 inhibitors has been considered a strategy to enhance CRISPR safety [95,96]. Thorough risk–benefit assessments are essential before applying CRISPR/Cas9 gene editing technology for selective gene therapy. Overcoming these limitations requires further research and development.

## 7. Conclusions

The CRISPR/Cas9 system, which was initially derived from bacterial adaptive immune systems, has revolutionized gene editing in the life sciences and provided a powerful tool for studying human tumors. As research in cancer treatment advances, addressing challenges in genome editing is crucial. This involves improving target specificity, reducing off-target effects, and solving the delivery issue of the CRISPR system.

Researchers have used in vivo or in vitro CRISPR screenings to uncover new tumor targets, but heterogeneity among tumors and patients when utilizing CRISPR technology for whole-genome screenings is crucial to consider: the more the constructed tumor model resembles the actual tumor phenotype, the more likely it is the results will provide actionable new targets suitable for related tumor types. Currently, PDX models are considered the most accurate tools for describing patient tumor characteristics. Expanding research on PDX models is essential to understand patient heterogeneity and tumor development mechanisms, guiding personalized therapies.

In the field of immune oncology, breakthroughs include promising results from autologous CAR-T-cell therapies in cancer treatment. However, challenges like high production costs can be addressed with universal CAR-T cells derived from healthy donors, modified using CRISPR gene editing for improved antitumor activity and lower adverse reactions. While the CRISPR/Cas9 system is rapidly moving towards its clinical potential, ongoing research and improvements are necessary for its safe and effective application in cancer treatment.

## Figures and Tables

**Figure 1 ijms-24-16325-f001:**
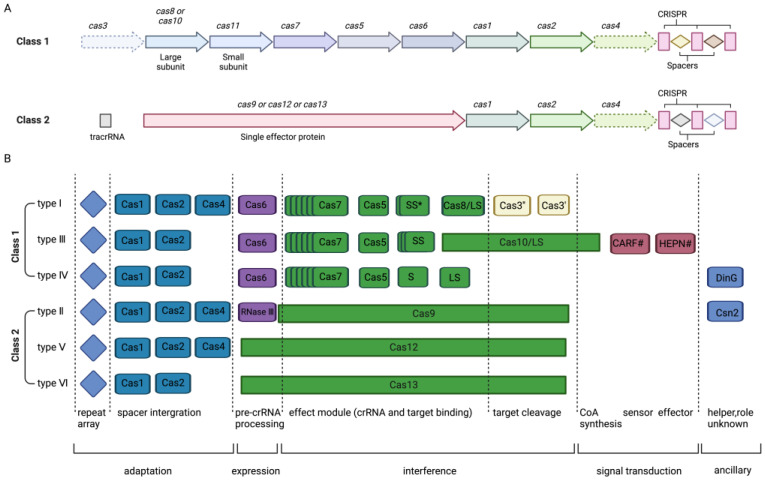
Schematic representation of the characteristic CRISPR-Cas loci for each class 1 and 2 subtype. Class 1 CRISPR-Cas systems are characterized by effector components composed of multiple Cas proteins. These proteins collectively form a complex capable of binding crRNA, working collaboratively in the binding and processing of the target. In contrast, class 2 systems are distinguished by possessing a single multidomain crRNA-binding protein functionally analogous to the entire effector assembly of class 1. (**A**) delineates the foundational architectural organization of both class 1 and class 2 CRISPR-Cas systems. (**B**) offers a comprehensive depiction of the functional modules inherent within the CRISPR-Cas systems. Cas3 is depicted as the fusion of two distinct genes: one encoding the helicase Cas3′ and the other responsible for the nuclease HD Cas3″, with Cas3″ often near cas3′. This illustration showcases the conventional associations amongst the genetic, structural, and functional organizations across the six variants of CRISPR-Cas systems. The nomenclature for the proteins adheres to contemporary conventions. The asterisk denotes potential small subunits that might fuse with the large subunit in certain subtypes of type I. Homologous genes are shown by the same color. Function descriptions corresponding to different colors and shapes are provided at the bottom of the figure, separated by dashed lines. The pound symbol (#) hints at the presence of other, yet-to-be-identified sensor, effector, and ring nuclease protein families potentially operating within the same signaling cascade.

**Figure 2 ijms-24-16325-f002:**
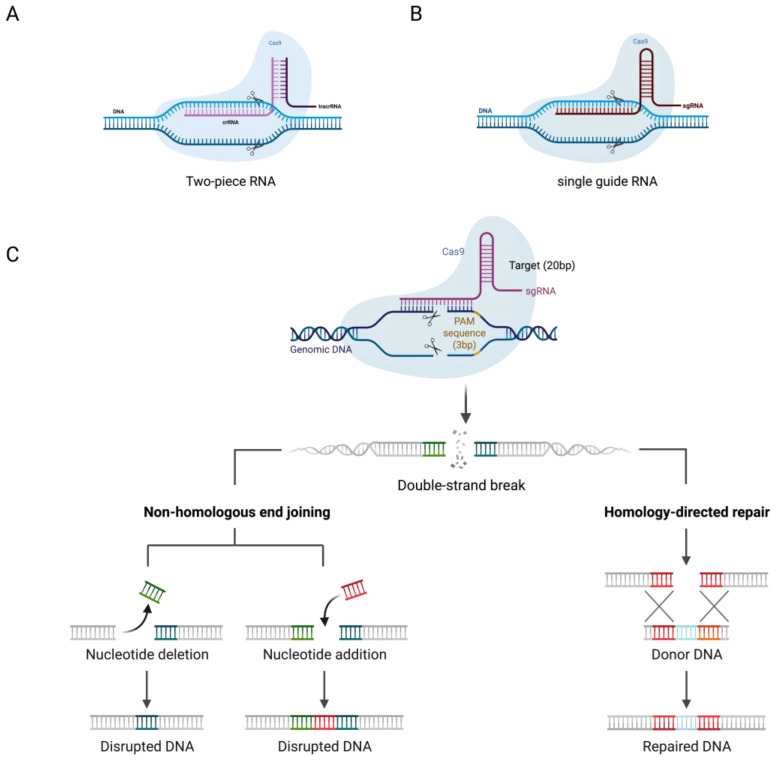
Mechanism of the CRISPR/Cas9 gene editing system. (**A**) The two-part guide RNA consists of a duplex of a tracrRNA and a crRNA. (**B**) Single-guide RNA (sgRNA) was created by fusing the crRNA and tracrRNA sequences together into a single RNA chimera by creating a loop at the end of the duplex region. (**C**) The single-guide RNA (sgRNA) directs the Cas9 nuclease to a complementary sequence in the genome, leading to DSBs. The CRISPR/Cas9 system comprises three components: an endonuclease (Cas9), CRISPR RNA (crRNA), and transactivating crRNA (tracrRNA). The guide RNA (gRNA) forms a duplex structure from crRNA and tracrRNA molecules. The sgRNA includes a unique 20-base pair (bp) sequence designed to complement the target DNA site, followed by a short DNA sequence known as the protospacer adjacent motif (PAM). The Cas9 nuclease is guided by the sgRNA to induce DSBs around the PAM region. DSBs are repaired through non-homologous end joining (NHEJ) or, in the presence of a DNA repair template, homology-directed repair (HDR). The latter can be exploited to introduce precise genetic modifications or exogenous sequences.

**Figure 3 ijms-24-16325-f003:**
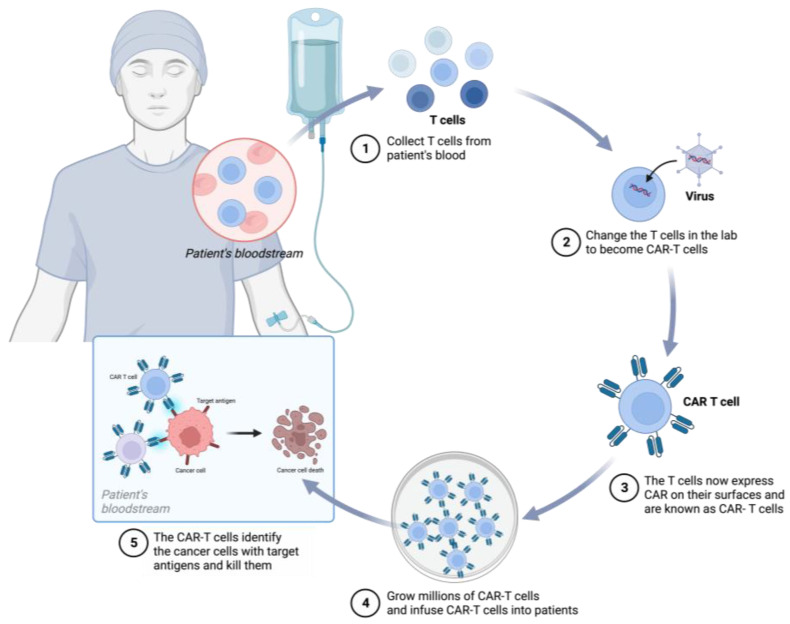
CAR-T cell therapy involves genetically engineering T cells in a laboratory to target specific proteins (antigens) in cancer cells and induce their destruction. (1) T cells are extracted from the patient’s blood. (2) The gene for a specialized receptor called a chimeric antigen receptor (CAR) is inserted into the T cells in the laboratory. This gene encodes an engineered CAR protein expressed on the surface of the T cells, creating CAR-T cells. (3) Millions of CAR-T cells are cultured in the laboratory. (4) Subsequently, CAR-T cells are administered intravenously to the patient. (5) CAR-T cells bind to antigens on cancer cells and kill them.

**Figure 4 ijms-24-16325-f004:**
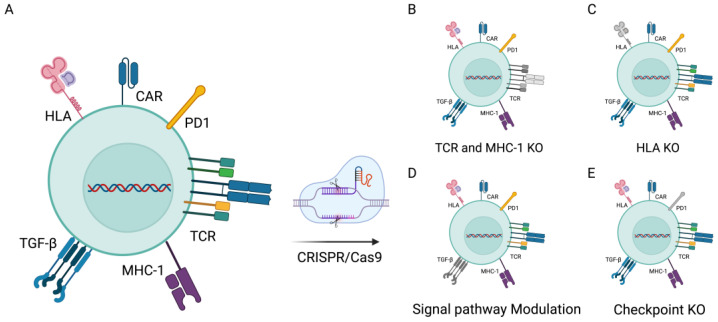
CRISPR/Cas9 gene editing in CAR-T cells (**A**) Extensive testing of gene knockouts in T cells. (**B**) Knocking out the TCR to reduce the likelihood of graft-versus-host disease (GvHD). (**C**) Knocking out HLAs to enhance the persistence of gene-modified cells. (**D**) Significant enhancement of CAR-T cells through the modulation of CAR-T-cell signaling via the inhibition of immunosuppressive TGF-β signaling. (**E**) Increased antitumor activity in T cells by knocking out T-cell checkpoint inhibitory receptors, such as PD-1.

**Figure 5 ijms-24-16325-f005:**
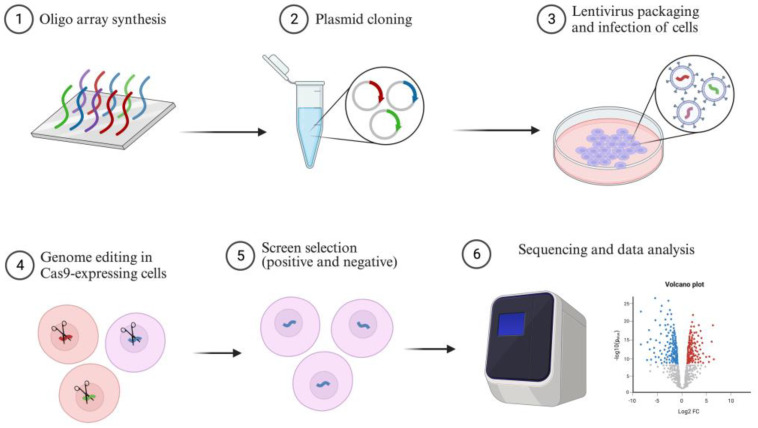
An overview of the CRISPR screening process. It begins with the construction of synthetic sgRNA plasmid libraries for Cas9 knockout or transcriptional activation. These plasmid libraries are then packaged into lentiviruses and transduced into the target cell lines, creating stable cell lines for screening. After selection, a diverse pool of cells with gene knockouts is obtained, which can be used in various screening methods. Genomic DNA is collected at specific time points, followed by next-generation sequencing and data analysis to identify candidate genes.

**Table 1 ijms-24-16325-t001:** Comparative overview of autologous and allogeneic CAR-T-cell therapies.

Feature	Autologous CAR-T-Cell Therapy	Allogeneic CAR-T-Cell Therapy
Definition	Patient’s own T cells	Donor T cells
Immunocompatibility	High	Moderate/Low
Preparation Time	Longer	Shorter
Cost	Higher	lower
GvHD Risks	Low	Moderate/High
Treatment Delay	Possible	Less
Indications	Most treatments	Specific cases
Efficacy	Variable	Higher
Cost Effectiveness	Moderate	Higher

**Table 2 ijms-24-16325-t002:** Recent clinical trials of CAR-T-cell therapies in different cancers.

CAR-T Cell	Tumor Type	Phase of Development	Country	Clinical Trial Identifier
αPD1-MSLN-CAR-T cells	Solid Tumor	Early Phase 1	China	NCT05373147
CD20/CD22 dual-targeted CAR-T cells	Lymphoid Hematological Malignancies	Early Phase 1	China	NCT04283006
bi-4SCAR CD19/79b CAR-T cells	B-Cell Malignancies	Phase 1	China	NCT05436509
CD5.CAR/28zeta CAR-T cells	T-Cell Malignancies	Phase 1	America	NCT03081910
CD19 CAR-T cells	Lymphoblastic B-Cell Lymphoma	Phase 1	Belarus	NCT05333302
GPC3 CAR-T cells	Hepatocellular Carcinoma	Phase 1	America	NCT05003895
CD44v6 CAR-T cells	Cancers Which Are CD44v6-Positive	Phase 2	China	NCT04427449
UCD19 CAR-T cells	B-cell Acute Lymphoblastic Leukemia; B-Cell Non-Hodgkin Lymphoma	Phase 2	America	NCT04544592

**Table 3 ijms-24-16325-t003:** Summary of CRISPR screening results in cancer research.

Target Carcinoma	Screening Purpose	CRISPR Screening Type	Experimental Setting	Candidate Gene	Literature Reference
Breast cancer	immune therapy	CRISPRKO	in vivo	*Cop1*	[58]
Bladder cancer	drug resistance	CRISPRKO	in vitro	*HNRNPU*	[59]
Gastric cancer	isoforms function	CRISPRi	in vitro	*ZFHX3*	[63]
Glioblastoma	lncRNA function	CRISPRi	in vitro	*lncGRS-1*	[65]
Acute myeloid leukemia	lncRNA	CRISPRa	in vitro/vivo	*GAS6-AS2*	[69]
Lung cancer	immune therapy	CRISPRa	in vitro	*IFNG*	[70]
Melanoma	immune therapy	CRISPRa	in vitro	*BCL-2/B3GNT2*	[71]

## Data Availability

Data are contained within the article.

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
