# Peer review of "CRISPR/Cas9-Mediated Genome Editing in Cancer Therapy"

_ijms, 2023, doi:10.3390/ijms242216325_

Round 1
Reviewer 1 Report
Comments and Suggestions for Authors
In this review, the authors summarize different aspects of cancer research and treatment that have benefitted, and can potentially benefit in the future, from the implementation of CRISPR-based genome editing technology. The subject matter is timely and interesting, and the choices of which specific topics to cover within the review were appropriate. Nevertheless, there are several issues that need to be addressed.
1. All of the figures were clearly developed using the “BioRender” tool, in several cases with barely any modification of the pre-made templates, yet no citation was provided.
2. The section in the Introduction where the CRISPR system is described is inaccurate and misleading (lines 45-53). The description of the natural CRISPR system is mixed in and confused with leverage of the CRISPR system as a technology for genome editing. For example,
Lines 47-48: This system combats viruses and plasmids, through a single guide RNA (sgRNA) and Cas9 nuclease.
The sgRNA is an artifical design to facilitate use of the system in genome engineering. The natural Cas9 has a two part RNA – the trRNA and crRNA.
Also, line 52 describes the intracellular repair systems that repair the double strand break created by Cas9. However, in the natural system, the CRISPR system cuts invading phage or plasmid DNA – with the goal of disabling it. I have not seen reports that the endogenous repair systems are engaged in this case. The endogenous repair systems come into play when the CRISPR system is used to enable genome engineering.
Therefore, this section requires rewriting.
3. In section 3, the authors describe CAR-T cells with an abundance of technical detail, but do not provide a focused description of how the current CAR-T technology benefits, or could potentially benefit, from CRISPR gene editing technology. A figure illustrating CAR-T cells and their designed receptors would be helpful. A distinction between CAR-T deriving from the patient’s own T-cells, versus universal donors, should be made. The advantages and disadvantages of each could be presented in a table.
4. The instances where CRISPR was used to improve CAR-T (such as deleting endogenous TCR) should be explained more fully, otherwise it is just a listing that is not very useful to the reader (lines 186-205). The molecular limitations of the original CAR-T should be properly described and explained, and then the CRISPR strategies used to solve the issue. The authors should describe whether CRISPR knock-out or knock-in was used, and what the issues are in implementing these in the clinic. As a side note, reference 15 should be included in this section. Reference 15 was used earlier when repair of DSBs by HDR was described. Although reference 15 reports on CRISPR with HDR-directed repair, used in the production of CAR-T cells, it is not necessarily a proper reference for the general statement about HDR (line 102).
5. The statement on line 293-4 is incorrect. “The sgRNA, typically spanning 50 base pairs upstream to 300 base pairs downstream of the transcription start site (TSS), consists of 102 nucleotides. It include a 20-nt target-specific complementary region, a 42-nt Cas9-binding RNA structure, and a 40-nt transcription termination sequence derived from S. pyogenes[5].” There is confusion of the genomic sites targeted in the CRISPRi strategy with the design of sgRNA.
6. Line 303. VP64 is an activator, and should not be included in the section on CRISPRi.
7. Line 327. To describe CRISPRa as “endogenous” activation is misleading, since the promoters are activated through CRISPRa using artificial activators, which are likely to produce levels and timing of gene expression that are completely different from the endogenous activation.
8. The description of the study by Wang (line 355) is not clear.
9. The study cited in reference 87 is misrepresented in line 548-9. This study raised the problem of p53 activation via CRIPSR-induced DSBs leading to apoptosis of targeted cells, and therefore risking enrichment for cells mutated in p53 in the surviving cell population.
Comments on the Quality of English LanguageThe English is generally good, but there are instances of mistakes and non-standard usage. For example:
line 68: "shed lights" (standard usage is "shed light")
Figure 2 (within figure) non-homologous is misspelled
line 136: "CAR-T cells genetically" should be "CAR-T cells are genetically"
line 138: "These process activities" should be "This process activates"
Note that this is not an exhaustive list. Please have an editor/proofreader check the entire manuscript.
Author Response
Reviewer 1 #Review Report(Round 1)
- All of the figures were clearly developed using the “BioRender” tool, in several cases with barely any modification of the pre-made templates, yet no citation was provided.
REPLY: Thank you for bringing this to our attention. We appreciate your understanding and constructive feedback. We have included a statement in the acknowledgments section of the revised manuscript to explicitly mention the use of the BioRender tool for creating the figures and to ensure proper citation in accordance with usage guidelines, thus upholding intellectual property rights and academic integrity in our work.
Line 682
- The section in the Introduction where the CRISPR system is described is inaccurate and misleading (lines 45-53). The description of the natural CRISPR system is mixed in and confused with leverage of the CRISPR system as a technology for genome editing. For example,
Lines 47-48: This system combats viruses and plasmids, through a single guide RNA (sgRNA) and Cas9 nuclease.
The sgRNA is an artifical design to facilitate use of the system in genome engineering. The natural Cas9 has a two part RNA – the trRNA and crRNA.
Also, line 52 describes the intracellular repair systems that repair the double strand break created by Cas9. However, in the natural system, the CRISPR system cuts invading phage or plasmid DNA – with the goal of disabling it. I have not seen reports that the endogenous repair systems are engaged in this case. The endogenous repair systems come into play when the CRISPR system is used to enable genome engineering.
Therefore, this section requires rewriting.
REPLY: Thank you for your detailed comments on the section of our manuscript that deals with the CRISPR system. We acknowledge the inaccuracies you highlighted regarding the natural function of CRISPR versus its adapted use in genome editing technologies. It was not our intention to conflate the two, and we recognized the critical importance of clearly distinguishing between the natural biological processes and the technological applications that have been developed from them.
In response to your feedback, we have carefully revised the Introduction to accurately reflect the dual nature of CRISPR: its role in adaptive bacterial immunity, distinguishing between the naturally occurring trRNA and crRNA components, and the engineered sgRNA utilized in CRISPR-based genome editing.
Line 48-64
- In section 3, the authors describe CAR-T cells with an abundance of technical detail, but do not provide a focused description of how the current CAR-T technology benefits, or could potentially benefit, from CRISPR gene editing technology. A figure illustrating CAR-T cells and their designed receptors would be helpful. A distinction between CAR-T deriving from the patient’s own T-cells, versus universal donors, should be made. The advantages and disadvantages of each could be presented in a table.
REPLY: Thank you for your insightful comments. We have included a focused description of how CRISPR gene editing technology benefits or has the potential to benefit current CAR-T cell therapies. An illustrative figure was added to visually depict CAR-T cells and their engineered receptors (Figure 4 on line 223). Furthermore, we clearly distinguished autologous CAR-T therapies, which use a patient’s own T-cells, and allogeneic CAR-T therapies from universal donors (Figure 4 legend on line 224-230). A table comparing the advantages and disadvantages of each approach was also included to provide a comprehensive overview (Table1 on line 231-232).
- The instances where CRISPR was used to improve CAR-T (such as deleting endogenous TCR) should be explained more fully, otherwise it is just a listing that is not very useful to the reader (lines 186-205). The molecular limitations of the original CAR-T should be properly described and explained, and then the CRISPR strategies used to solve the issue. The authors should describe whether CRISPR knock-out or knock-in was used, and what the issues are in implementing these in the clinic. As a side note, reference 15 should be included in this section. Reference 15 was used earlier when repair of DSBs by HDR was described. Although reference 15 reports on CRISPR with HDR-directed repair, used in the production of CAR-T cells, it is not necessarily a proper reference for the general statement about HDR (line 102).
REPLY: Thank you for your valuable feedback. We understand the importance of providing a comprehensive explanation on the use of CRISPR to enhance CAR-T cell technology.
We have extended the discussion on the instances where CRISPR was employed to improve CAR-T cells, including a more thorough explanation of the deletion of endogenous TCR, among other modifications. We also described and explained the molecular limitations of the original CAR-T technology in more detail, followed by how CRISPR strategies were employed to overcome these limitations. We also specified whether CRISPR knock-out or knock-in techniques were used, and discuss the challenges and considerations in translating these techniques to the clinic.
We have included Reference 15 in this section as suggested, and re-evaluate its placement in line 102 regarding the general statement about HDR.
We appreciate your meticulous review and constructive suggestions which will undoubtedly improve the quality of our manuscript.
Line 234-274
- The statement on line 293-4 is incorrect. “The sgRNA, typically spanning 50 basepairs upstream to 300 base pairs downstream of the transcription start site (TSS),consists of 102 nucleotides. It include a 20-nt target-specific complementary region, a 42-nt Cas9 binding RNA structure, and a 40-nt transcription termination sequencederived from S. pyogenes[5].” There is confusion of the genomic sites targeted in the CRISPRi strategy with the design of sgRNA.
REPLY: Thank you for pointing out the inaccuracies in the statement on lines 293-4. We understand the confusion and misrepresentation regarding the CRISPRi strategy and the design of sgRNA. We have corrected this information to provide a precise and accurate description of the sgRNA design and its role in the CRISPRi strategy.
Line 375-379
- Line 303. VP64 is an activator, and should not be included in the section on CRISPRi.
REPLY: Thank you for your meticulous observation on line 303 regarding the inclusion of VP64, an activator, in the section on CRISPRi. We acknowledge the oversight and understand that VP64 should not have been included in this section given its activation function rather than inhibition. We have rectified this error by removing the mention of VP64 from the CRISPRi section and ensure the context is accurately presented in the revised manuscript.
Line 386
- Line 327. To describe CRISPRa as “endogenous” activation is misleading, since the promoters are activated through CRISPRa using artificial activators, which are likely to produce levels and timing of gene expression that are completely different from the endogenous activation.
REPLY: Thank you for bringing to our attention the misleading terminology used in line 327 to describe CRISPRa as "endogenous" activation. We understand the distinction you are drawing attention to, and agree that the use of artificial activators in CRISPRa indeed results in a modulation of gene expression that may diverge significantly from endogenous activation patterns.
We have amended the terminology in the mentioned line to accurately reflect the mechanism of CRISPRa, and ensure clarity in conveying that the activation achieved through CRISPRa is facilitated by artificial activators, which may lead to gene expression levels and timing distinct from natural, endogenous activation. We appreciate your keen observation and guidance, which contribute to improving the clarity and accuracy of our manuscript.
Line 409-413
- The description of the study by Wang (line 355) is not clear.
REPLY: Thank you for pointing out the lack of clarity in the description of the study by Wang on line 355. We understand that the details provided may not sufficient or clear enough for readers to grasp the key findings and significance of the study. To address this, we have revised it to provide a more clear and detailed description of Wang's study, ensuring to highlight the main findings, methodologies, and the relevance of the study in the context of our manuscript.
Line 438-453
- The study cited in reference 87 is misrepresented in line 548-9. This study raised the problem of p53 activation via CRIPSR-induced DSBs leading to apoptosis of targeted cells, and therefore risking enrichment for cells mutated in p53 in the surviving cell population.
REPLY: Thank you for bringing to our attention the misrepresentation of the study cited in reference 87 on line 548-9. We understand the critical distinction regarding the problem of p53 activation and its implications as outlined in the mentioned study.
To address this, we revised the text on lines 548-9 to accurately reflect the findings and concerns raised in the cited study regarding CRISPR-induced double-strand breaks (DSBs) leading to p53 activation, apoptosis of targeted cells, and the consequent risk of enriching for cells mutated in p53 in the surviving cell population.
Line 646-651
Reviewer 2 Report
Comments and Suggestions for Authors
This paper is a very important Review that covers a wide range of topics from an overview of CRISPR/Cas, its application in cancer research, and the application of CAR-T cells and CRISPR in combination with PDX models. This Review will help cancer research using CRISPR. However, the following points need to be revised.
Major revision
・In Figure 1, the depiction of Type 1-6 CRISPR and Cas1-13 is unclear what the figure means for. Please also elaborate on the Figure legend that explains it. Since there are many explanations about Cas9 in the later, it may not be necessary to explain the above.
・In CRISPR/Cas9 in CAR-T cell therapies, CAR T-cells are created from Cas9-transfected T-cells, but is there any concern about Cas9? Cas9 is a gene derived from bacteria that does not exist in humans, and is there any effect when this gene is returned to the human body? These need to be explained.
・Is Table 1 shows clinical trials of CAR-T cells using CRISPR technology? If so, it is necessary to describe what molecules are knocked out by CRISPR/Cas.
・The "4 CRISPR screening in cancer" and "5 CRISPR/Cas9 in cancer model" overlap in content. I suggest that “4 CRISPR screening in cancer" only introduces CRISPR KO screening, CRISPRi screening, and CRISPRa screening, while the research results are introduced in "5 CRISPR/Cas9 in cancer model".
・In "4 CRISPR screening in cancer" and "5 CRISPR/Cas9 in cancer model," the results of CRISPR screening should be summarized in a table. The results of CRISPR screening are summarized in a table for easy reading and understanding. It would be excellent if the target carcinoma, purpose of screening, which types of CRISPR screenin, in vitro or in vivo, candidate gene, literature number, etc. are described.
Minor revision
・There is a period before the reference number [5] in Line 53.
・If the function of TRBC in Line 188 is similar to that of TRAC in Line 165, please explain them together.
・Line227, reference 36, if the Human genome project is milestone, the reference on 2023 seems inappropriate. https://www.nature.com/articles/35057062 or https://www. science.org/lookup/doi/10.1126/science.1058040 would be appropriate to cite.
Comments on the Quality of English Language
With the exception of the Minor revision item, the English is accurate.
Author Response
Reviewer 2 #Review Report(Round 1)
1・In Figure 1, the depiction of Type 1-6 CRISPR and Cas1-13 is unclear what the figure means for. Please also elaborate on the Figure legend that explains it. Since there are many explanations about Cas9 in the later, it may not be necessary to explain the above.
REPLY: Thank you for your feedback on Figure 1 and its accompanying legend. We understand that the depiction of Type 1-6 CRISPR and Cas1-13 may not be clear, and the intention behind including this figure may not have been effectively communicated.
To address your concerns, we revised Figure 1 to ensure that the depiction of Type 1-6 CRISPR and Cas1-13 is clear and informative. We also revised the figure legend to provide a more detailed and clear explanation of the figure, ensuring that it succinctly conveys the intended message and complements the information provided in the figure. Line 87-100
We greatly value your input, which will undoubtedly help in enhancing the clarity and coherence of our manuscript.
- 2. In CRISPR/Cas9 in CAR-T cell therapies, CAR T-cells are created from Cas9-transfected T-cells, but is there any concern about Cas9? Cas9 is a gene derived from bacteria that does not exist in humans, and is there any effect when this gene is returned to the human body? These need to be explained.
REPLY: Thank you for your insightful observations regarding the application of CRISPR/Cas9 technology in CAR-T cell therapies. Your concern about the potential immunogenicity of the bacterial-derived Cas9 protein is indeed a significant one. As elucidated in your comments, the immune response triggered by the exogenous nature of Cas9 could undermine the effectiveness of the gene-editing process, particularly in therapies requiring prolonged expression of Cas9.
In accordance with your suggestions, we have elaborated on the immunogenic risks associated with Cas9 and the innovative strategies that developed to mitigate these risks. These include the utilization of modified Cas9 variants to minimize immunogenic epitopes, transient expression systems, and the use of non-inflammatory vectors like adeno-associated viruses (AAV) to reduce immune activation. By incorporating these advancements, we aim to highlight the ongoing efforts to enhance the safety and efficacy of CRISPR/Cas9-based CAR-T cell therapies. Line 276-288
- 3. Is Table 1 shows clinical trials of CAR-T cells using CRISPR technology? If so, it is necessary to describe what molecules are knocked out by CRISPR/Cas.
REPLY: Thank you again for your attention to our work. Regarding the table of clinical trials towards CAR-T cells, actually we don’t know which molecules are knocked out by CRISPR/Cas in the CAR-T cells since the information from the clinical trials currently were not provided. We only presented the antigen targeted by CAR-T cells which were shown in Column 1 (CAR-T cell).
Previous is Table 1, current is Table 2.
- 4. The "4 CRISPR screening in cancer" and "5 CRISPR/Cas9 in cancer model" overlap in content. I suggest that “4 CRISPR screening in cancer" only introduces CRISPR KO screening, CRISPRi screening, and CRISPRa screening, while the research results are introduced in "5 CRISPR/Cas9 in cancer model".
REPLY: Thank you for your valuable feedback. Regarding the overlap between Section 4 "Applications of CRISPR Screening in Cancer" and Section 5 "CRISPR/Cas9 in Cancer Models", we have given this careful consideration. We recognize that there is an intersection between the two sections, but our aim is to distinguish between the detailed introduction of screening technologies (Section 4) and the specific research findings from the application of these technologies in particular cancer models (Section 5).
Section 4 focuses on the CRISPR screening technologies themselves, including principles and protocols of knockout (KO), interference (CRISPRi), and activation (CRISPRa) screens, with the aim to provide readers with sufficient background information to understand the broad prospects of these technologies in cancer research.
Section 5, on the other hand, looks at the application of these screening technologies in specific cancer models, emphasizing the new insights gained through the use of these technologies, and how they can advance our understanding of cancer pathogenesis and the development of treatment strategies.
We believe that this division more effectively communicates the unique value of each section and provides a clear informational hierarchy for readers. We wish to maintain the current structure and add directional statements in appropriate sections to minimize redundancy and strengthen the connection between sections.
We look forward to further guidance and will continue to work on improving our manuscript.
- 5. In "4 CRISPR screening in cancer" and "5 CRISPR/Cas9 in cancer model," the results of CRISPR screening should be summarized in a table. The results of CRISPR screening are summarized in a table for easy reading and understanding. It would be excellent if the target carcinoma, purpose of screening, which types of CRISPR screening, in vitro or in vivo, candidate gene, literature number, etc. are described.
REPLY: Thank you for your valuable suggestion. Regarding sections "4 CRISPR screening in cancer" and "5 CRISPR/Cas9 in cancer model," we compiled the results of CRISPR screening into a table, as recommended. This table described the target cancer types, the purpose of screening, the types of screening (in vitro or in vivo), candidate genes, and literature references. We hope this will enhance the readability and understanding of the article, and effectively present our findings.
Lines 457-467
6.There is a period before the reference number [5] in Line 53.
REPLY: Thank you for pointing out the punctuation error. We have corrected it by removing the period before the reference number [5] in Line 64.
Previous Line is 53, current line is 64.
- 7. If the function of TRBC in Line 188 is similar to that of TRAC in Line 165, please explain them together.
REPLY: We appreciate your guidance. In response to the parallel functions of TRBC in Line 188 and TRAC in Line 165, we merged their discussions in the revised manuscript for a unified and comprehensive explanation.
Previous Line is 165-188, current line is 235-252.
- 8. Line227, reference 36, if the Human genome project is milestone, the reference on 2023 seems inappropriate. https://www.nature.com/articles/35057062 or https://www. science.org/lookup/doi/10.1126/science.1058040 would be appropriate to cite.
REPLY: We acknowledge the discrepancy and thank you for your attentiveness. According to your suggestions, we corrected the citations for the Human Genome Project in Line 308-309 to make the reference more suitable.
Previous Line is 227, current line is 308-309.
Reviewer 3 Report
Comments and Suggestions for Authors
Shuai Ding and colleagues present the manuscript entitled “CRISPR/Cas mediated genome editing in cancer therapy” where they provided a review on CRISPR technology and its application in cancer research. The paper is interesting and well organized in fact, it requires minor changes.
MINOR COMMENTS:
1. Line 1: The authors should provide references on the role of CRISPR/Cas 9 on outcome in cancer research (liver, breast, colon etc.)
2. Figure 2: Authors should improve this Figure. The mechanism described in the legend is not well represented.
Author Response
Reviewer 3#Review Report(Round 1)
- Line 1: The authors should provide references on the role of CRISPR/Cas 9 on outcome in cancer research (liver, breast, colon etc.)
REPLY: Thank you for pointing out the need for additional references. We have included relevant citations on the role of CRISPR/Cas9 in cancer research outcomes across various cancer types such as liver, breast, and colon cancer in our revised manuscript. These references will help underscore the significance of CRISPR/Cas9 in oncology.
Line 70
- Figure 2: Authors should improve this Figure. The mechanism described in the legend is not well represented.
REPLY: We appreciate your feedback on Figure 2. We have revised the figure to more accurately depict the mechanism outlined in the legend, ensuring that the visual representation clearly aligns with the described processes.
Lines 125-138
Round 2
Reviewer 2 Report
Comments and Suggestions for Authors
Dear Authors
Thank you very much for the revision in a short period of time.
Your corrections to the points I pointed out were also accurate, and your replies were also appropriate.
There are three minor revisions that need to be made.
・In Table 3 Experimental Setting, in vitro, in vivo and Gene name of Candidate Gene should be italicized.
・References 39 and 40 are duplicated.
・Reference 43, 44 and 47 have too many authors' names. Please follow the submission rules.
Comments on the Quality of English LanguageEnglish was appropriate except for mentioned in "Comments and Suggestions for Authors
"
Author Response
Reviewer 2 #Review Report (Round 2)
1.In Table 3 Experimental Setting, in vitro, in vivo and Gene name of Candidate Gene should be italicized.
REPLY: Thank you for your careful reading of our manuscript and for your suggestion regarding the formatting of Table 3. We have revised Table 3 accordingly to reflect these changes. Line 467
2.References 39 and 40 are duplicated.
REPLY: We have addressed the errors identified in our reference list. The duplicated references 39 and 40 have been corrected by removing the repetition and re-citing the appropriate literature. Line 286
3.Reference 43, 44 and 47 have too many authors' names. Please follow the submission rules.
REPLY: Thank you for your comments and guidance on our reference list. We have revised the references to comply with the journal's submission rules. Line 780,782, 789